# Discrete Autoencoders for Sequence Models

## Abstract

Recurrent models for sequences have been recently successful at many tasks, especially for language modeling and machine translation. Nevertheless, it remains challenging to extract good representations from these models. For instance, even though language has a clear hierarchical structure going from characters through words to sentences, it is not apparent in current language models. We propose to improve the representation in sequence models by augmenting current approaches with an autoencoder that is forced to compress the sequence through an intermediate discrete latent space. In order to propagate gradients though this discrete representation we introduce an improved semantic hashing technique. We show that this technique performs well on a newly proposed quantitative efficiency measure. We also analyze latent codes produced by the model showing how they correspond to words and phrases. Finally, we present an application of the autoencoder-augmented model to generating diverse translations.

## 1 Introduction

Autoencoders have a long history in deep learning (Hinton & Salakhutdinov, 2006; Salakhutdinov & Hinton, 2009a; Vincent et al., 2010; Kingma & Welling, 2013). In most cases, autoencoders operate on continuous representations, either by simply making a bottleneck (Hinton & Salakhutdinov, 2006), denoising (Vincent et al., 2010), or adding a variational component (Kingma & Welling, 2013). In many cases though, a discrete latent representation is potentially a better fit.

Language is inherently discrete, and autoregressive models based on sequences of discrete symbols yield impressive results. A discrete representation can be fed into a reasoning or planning system or act as a bridge towards any other part of a larger system. Even in reinforcement learning where action spaces are naturally continuous, Metz et al. (2017) show that discretizing them and using autoregressive models can yield improvements.

Unluckily, using discrete latent variables is challenging in deep learning. And even with continuous autoencoders, the interactions with an autoregressive component cause difficulties. Despite some success (Bowman et al., 2016; Yang et al., 2017), the task of meaningfully autoencoding text in the presence of an autoregressive decoder has remained a challenge.

In this work we present an architecture that autoencodes a sequence $s$ of $N$ discrete symbols from any vocabulary (e.g., a tokenized sentence), into a $K$-fold (we test $K = 8$ and $K = 32$) compressed sequence $c(s)$ of $\lceil \frac{N}{K} \rceil$ latent symbols from a new vocabulary which is learned. The compressed sequence is generated to minimize perplexity in a (possibly conditional) language model trained to predict the next token on $c(s) \circ s$: the concatenation of $c(s)$ with the original sequence $s$.

Since gradient signals can vanish when propagating over discrete variables, the compression function $c(s)$ can be hard to train. To solve this problem, we draw from the old technique of *semantic hashing* (Salakhutdinov & Hinton, 2009b). There, to discretize a dense vector $v$ one computes $\sigma(v + n)$ where $\sigma$ is the sigmoid function and $n$ represents annealed Gaussian noise that pushes the network to not use middle values in $v$. We enhance this method by using a saturating sigmoid and a straight-through pass with only bits passed forward. These techniques, described in detail below, allow to forgo the annealing of the noise and provide a stable discretization mechanism that requires neither annealing nor additional loss factors.

We test our discretization technique by amending language models over $s$ with the autoencoded sequence $c(s)$. We compare the perplexity achieved on $s$ with and without the $c(s)$ component,

and contrast this value with the number of bits used in $c(s)$. We argue that this number is a proper measure for the performance of a discrete autoencoder. It is easy to compute and captures the performance of the autoencoding part of the model. This quantitative measure allows us to compare the technique we introduce with other methods, and we show that it performs better than a Gumbel-Softmax (Jang et al., 2016; Maddison et al., 2016) in this context.

Finally, we discuss the use of adding the autoencoded part $c(s)$ to a sequence model. We present samples from a character-level language model and show that the latent symbols correspond to words and phrases when the architecture of $c(s)$ is local. ehen, we introduce a decoding method in which $c(s)$ is sampled and then $s$ is decoded using beam search. This method alleviates a number of problems observed with beam search or pure sampling. We show how our decoding method can be used to obtain diverse translations of a sentence from a neural machine translation model. To summarize, the main contributions of this paper are:

(1) a discretization technique that works well without any extra losses or parameters to tune,

(2) a way to measure performance of autoencoders for sequence models with baselines,

(3) an improved way to sample from sequence models trained with an autoencoder part.

## 2 TECHNIQUES

Below, we introduce our discretization method, the autoencoding function $c(s)$ and finally the complete model that we use for our experiments. All code and hyperparameter settings needed to replicate our experiments will be available as open-source[1].

### 2.1 DISCRETIZATION BY IMPROVED SEMANTIC HASHING

As already mentioned above, our discretization method stems from semantic hashing (Salakhutdinov & Hinton, 2009b). To discretize a $b$-dimensional vector $v$, we first add noise, so $v^n = v + n$. The noise $n$ is drawn from a $b$-dimensional Gaussian distribution with mean 0 and standard deviation 1 (deviations between 0 and 1.5 all work fine, see ablations below). The sum is component-wise, as are all operations below. Note that noise is used only for training, during evaluation and inference $n = 0$. From $v^n$ we compute two vectors: $v_1 = \sigma'(v^n)$ and $v_2 = (v^n < 0)$, where $\sigma'$ is the saturating sigmoid function from (Kaiser & Sutskever, 2016; Kaiser & Bengio, 2016):

$$\sigma'(x) = \max(0, \min(1, 1.2\sigma(x) - 0.1)).$$

The vector $v_2$ represents the discretized value of $v$ and is used for evaluation and inference. During training, in the forward pass we use $v_1$ half of the time and $v_2$ the other half. In the backward pass, we let gradients always flow to $v_1$, even if we used $v_2$ in the forward computation[2].

We will denote the vector $v$ discretized in the above way by $v^d$. Note that if $v$ is $b$-dimensional then $v^d$ will have $b$ bits. Since in other parts of the system we will predict $v^d$ with a softmax, we want the number of bits to not be too large. In our experiments we stick with $b = 16$, so $v^d$ is a vector of 16 bits, and so can be interpreted as an integer between 0 and $2^{16} - 1 = 65535$.

The dense vectors representing activations in our sequence models have much larger dimensionality than 16 (often 512, see the details in the experimental section below). To discretize such a high-dimensional vector $w$ we first have a simple fully-connected layer converting it into $v = \text{dense}(w, 16)$. In our notation, $\text{dense}(x, n)$ denotes a fully-connected layer applied to $x$ and mapping it into $n$ dimensions, i.e., $\text{dense}(x, n) = xW + B$ where $W$ is a learned matrix of shape $d \times n$, where $d$ is the dimensionality of $x$, and $B$ is a learned bias vector of size $n$. The discretized vector $v^d$ is converted back into a high-dimensional vector using a 3-layer feed-forward network:

```
h1a = dense(vd, filter_size)
h1b = dense(1.0 - vd, filter_size)
h2 = dense(relu(h1a + h1b), filter_size)
result = dense(relu(h2), hidden_size)
```

---

[1] Code available at redacted-for-blind-review

[2] This can be done in TensorFlow using: v₂ += v₁ - tf.stop_gradient(v₁).

Above, every time we apply `dense` we create a new weight matrix an bias to be learned. The `relu` function is defined in the standard way: $\mathrm{relu}(x) = \max(x, 0)$. In the network above, we usually use a large `filter_size`; in our experiments we set it to 4096 while `hidden_size` was usually 512. We suspect that this allows the above network to recover from the discretization bottleneck by simulating the distribution of $w$ encountered during training. Given a dense, high-dimensional vector $w$ we will denote the corresponding `result` returned from the network above by $\mathrm{bottleneck}(w)$ and the corresponding discrete vector $v_2$ by $\mathrm{discrete}(w)$.

## 2.2 GUMBEL-SOFTMAX FOR DISCRETIZATION

As an alternative discretization method, we consider the recently studied Gumbel-Softmax (Jang et al., 2016; Maddison et al., 2016). In that case, given a vector $w$ we compute $\mathrm{discrete}_g(w)$ by applying a linear layer mapping into $2^{16}$ elements, resulting in the logits $l$. During evaluation and inference we simply pick the index of $l$ with maximum value for $\mathrm{discrete}_g(w)$ and the vector $\mathrm{bottleneck}_g(w)$ is computed by an embedding. During training we first draw samples $g$ from the Gumbel distribution: $g \sim -\log(-\log(u))$, where $u \sim \mathcal{U}(0, 1)$ are uniform samples. Then, as in (Jang et al., 2016), we compute $x$, the log-softmax of $l$, and set:

$$y_i = \frac{\exp((x_i + g_i)/\tau)}{\sum_i \exp((x_i + g_i)/\tau)}.$$

With low temperature $\tau$ this vector is close to the 1-hot vector representing the maximum index of $l$. But with higher temperature, it is an approximation (see Figure 1 in Jang et al. (2016)). We multiply this vector $y$ by the embedding matrix to compute $\mathrm{bottleneck}_g(w)$ during training.

## 2.3 AUTOENCODING FUNCTION

Having the functions $\mathrm{bottleneck}(w)$ and $\mathrm{discrete}(w)$ (respectively their Gumbel-Softmax versions), we can now describe the architecture of the autoencoding function $c(s)$. We assume that $s$ is already a sequence of dense vectors, e.g., coming from embedding vectors from a tokenized sentence. To halve the size of $s$, we first apply to it 3 layers of 1-dimensional convolutions with kernel size 3 and padding with 0s on both sides (`SAME`-padding). We use ReLU non-linearities between the layers and layer-normalization (Ba et al., 2016). Then, we add the input to the result, forming a residual block. Finally, we process the result with a convolution with kernel size 2 and stride 2, effectively halving the size of $s$. In the *local* version of this function we only do the final strided convolution, without the residual block.

To autoencode a sequence $s$ and shorten it $K$-fold, with $K = 2^k$, we first apply the above step $k$ times obtaining a sequence $s'$ that is $K$ times shorter. Then we put it through the discretization bottleneck described above. The final compression function is given by $c(s) = \mathrm{bottleneck}(s')$ and the architecture described above is depicted in Figure 1.

Note that, since we perform 3 convolutions with kernel 3 in each step, the network has access to a large context: $3 \cdot 2^{k-1}$ just from the receptive fields of convolutions in the last step. That's why we also consider the local version. With only strided convolutions, the $i$-th symbol in the local $c(s)$ has only access to a fixed $2^k$ symbols from the sequence $s$ and can only compress them.

Training with $c(s)$ defined above from scratch is hard, since at the beginning of training $s'$ is generated by many layers of untrained convolutions that are only getting gradients through the discretization bottleneck. To help training, we add a side-path for $c(s)$ without discretization: we just use $c(s) = s'$ for the first 10000 training steps. In this pretraining stage the network reaches loss of almost 0 as everything needed to reconstruct $s$ is encoded in $s'$. After switching to $c(s) = \mathrm{bottleneck}(s')$ the loss is high again and improves during further training.

## 2.4 AUTOENCODING SEQUENCE MODEL

To test the autoencoding function $c(s)$ we will use it to prefix the sequence $s$ in a sequence model. Normally, a sequence model would generate the $i$-th element of $s$ conditioning on all elements of $s$ before that, $s_{<i}$, and possibly on some other inputs. For example, a language model would just condition on $s_{<i}$ while a neural machine translation model would condition on the input sentence (in the other language) and $s_{<i}$. We do not change the sequence models in any way other than adding

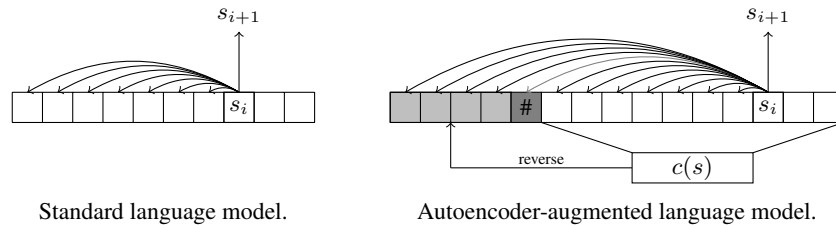

Figure 1: Architecture of the autoencoding function $c(s)$. We write $\texttt{conv}_{s=b}^{k=a}$ to denote a 1D convolutional layer with kernel size $a$ and stride $b$. See text for more details.

Standard language model.   Autoencoder-augmented language model.

Figure 2: Comparison of a standard language model and our autoencoder-augmented model. The architecture for $c(s)$ is presented in Figure 1 and the arrows from $s_i$ to $s_{<i}$ depict dependence.

the sequence $c(s)$ as the prefix of $s$. Actually, for reasons analogous to those in (Sutskever et al., 2014), we first reverse the sequence $c(s)$, then add a separator symbol (#), and only then concatenate it with $s$, as depicted in Figure 2. We also use a separate set of parameters for the model predicting $c(s)$ so as to make sure that the models predicting $s$ with and without $c(s)$ have the same capacity.

As the architecture for the sequence model we use the Transformer (Vaswani et al., 2017). Transformer is based on multiple attention layers and was originally introduced in the context of neural machine translation. We focused on the autoencoding function $c(s)$ and did not tune the sequence model in this work: we used all the defaults from the baseline provided by the Transformer authors (6 layers, hidden size of $512$ and filter size of $4096$) and only varied parameters relevant to $c(s)$.

## 3 EXPERIMENTS

We experimented with autoencoding on 3 different sequence tasks: (1) on a character-level language model, (2) on a word-level language model, and (3) on a word-level translation model. The goal for (1) was to check if our technique works at all, since character sequences are naturally amenable to compression into shorter sequences of objects from a larger vocabulary. For (2), we wanted to check if the good results obtained in (1) will still hold if the input is from a larger vocabulary and inherently more compressed space. Finally, in (3) we want to check if this method is applicable to conditional models and how it can be used to improve decoding.

We use the LM1B corpus (Chelba et al., 2013) for language modelling and we tokenize it using a subword (wordpiece) tokenizer (Sennrich et al., 2016) into a vocabulary of 32000 words and word-pieces. For translation, we use the WMT English-German corpus, similarly tokenized into a vocabulary of 32000 words and word-pieces[3].

Below we report both qualitative and quantitative results. First, we focus on measuring the performance of our autoencoder quantitatively. To do that, we introduce a measure of discrete autoen-

---

[3]We used `https://github.com/tensorflow/tensor2tensor` for data preparation.

| Problem | ln(p) | ln(p') | K | DSAE |
|---|---|---|---|---|
| LM-en (characters) | 1.027 | 0.822 | 32 | 59% |
| LM-en (word) | 3.586 | 2.823 | 8 | 55% |
| NMT-en-de (word) | 1.449 | 1.191 | 8 | 19% |
| LM-en (word, Gumbel-Softmax) | 3.586 | 3.417 | 8 | 12% |
| NMT-en-de (word, Gumbel-Softmax) | 1.449 | 1.512 | 8 | 0% |

Table 1: Log-perplexities per word of sequence models with and without autoencoders, and their autoencoding efficiency. Results for Gumbel-Softmax heavily depend on tuning; see text for details.

coder performance on sequence tasks and compare our semantic hashing based method to Gumbel-Softmax on this scale.

## 3.1 DISCRETE SEQUENCE AUTOENCODING EFFICIENCY

Sequence models trained for next-symbol prediction are usually trained (and often also evaluated) based on the *perplexity* per token that they reach. Perplexity is defined as $2^H$, where $H$ is the entropy (in bits) of a distribution. Therefore, a language model that reaches a per-word perplexity of $p$, say $p = 32$, on a sentence $s$ can be said to compress each word from $s$ into $\log(p) = 5$ bits of information.

Let us now assume that this model is allowed to access some additional bits of information about $s$ before decoding. In our autoencoding case, we let it peek at $c(s)$ before decoding $s$, and $c(s)$ has $K = 8$ times less symbols and $b = 16$ bits in each symbol. So $c(s)$ has the information capacity of 2 bits per word. If our autoencoder was perfectly aligned with the needs of the language model, then allowing it to peek into $c(s)$ would lower its information needs by these 2 bits per word. The perplexity $p'$ of the model with access to $c(s)$ would thus satisfy $\log_2(p') = 5 - 2 = 3$, so its perplexity would be $p' = 8$.

Getting the autoencoder $c(s)$ perfectly aligned with the language model is hard, so in practice the perplexity $p'$ is always higher. But since we measure it (and optimize for it during training), we can calculate how many bits has the $c(s)$ part actually contributed to lowering the perplexity. We calculate $\log_2(p) - \log_2(p')$ and then, if $c(s)$ is $K$-times shorter than $s$ and uses $b$ bits, we define the *discrete sequence autoencoding efficiency* as:

$$\text{DSAE} = \frac{K(\log_2(p) - \log_2(p'))}{b} = \frac{K(\ln(p) - \ln(p'))}{b \ln(2)}.$$

The second formulation is useful when the raw numbers are given as natural logarithms, as is often the case during neural networks training.

Defined in this way, DSAE measures how many of the available bits in $c(s)$ are actually used well by the model that peeks into the autoencoded part. Note that some models may have autoencoding capacity higher than the number of bits per word that $\log(p)$ indicates. In that case achieving DSAE=1 is impossible even if $\log(p') = 0$ and the autoencoding is perfect. One should be careful when reporting DSAE for such over-capacitated models.

So how does our method perform on DSAE and how does it compare with Gumbel-Softmax? In Table 1 we list log-perplexties of baseline and autoencoder models. We report numbers for the global version of $c(s)$ on our 3 problems and compare it to Gumbel-Softmax on word-level problems. We did not manage to run the Gumbel-Softmax on character-level data in our baseline configuration because it requires too much memory (as it needs to learn the embeddings for each latent discrete symbol). Also, we found that the results for Gumbel-Softmax heavily depend on how the temperature parameter $\tau$ is annealed during training. We tuned this on 5 runs of a smaller model and chose the best configuration. This was still not enough, as in many runs the Gumbel-Softmax would only utilize a small portion of the discrete symbols. We added an extra loss term to increase the variance of the Gumbel-Softmax and ran another 5 tuning runs to optimize this loss term. We used the best configuration for the experiments above. Still, we did not manage to get any information autoencoded in the translation model, and got only 12% efficiency in the language model (see Table 1).

| Noise standard deviation | ln(p) | ln(p') | K | DSAE |
|---|---|---|---|---|
| 1.5 | 3.912 | 3.313 | 8 | 43.2% |
| 1.0 | 3.912 | 3.239 | 8 | 48.5% |
| 0.5 | 3.912 | 3.236 | 8 | 48.5% |
| 0.0 | 3.912 | 3.288 | 8 | 45.0% |

Table 2: Autoencoder-augmented language models with different noise deviations. All values from no noise (0.0) upto a deviation of 1.5 yield DSAE between 40% and 50%.

Our method, on the other hand, was most efficient on character-level language modeling, where we reach almost 60% efficiency, and it retained high 55% efficiency on the word-level language modeling task. On the translation task, our efficiency goes down to 19%, possibly because the $c(s)$ function does not take inputs into account, and so may not be able to compress the right parts to align with the conditional model that outputs $s$ depending on the inputs. But even with 19% efficiency it is still useful for sampling from the model, as shown below.

## 3.2 SENSITIVITY TO NOISE

To make sure that our autoencoding method is stable, we experiment with different standard deviations for the noise $n$ in the semantic hashing part. We perform these experiments on word-level language modelling with a smaller model configuration (3 layers, hidden size of 384 and filter size of 2048). The results, presented in Table 2, show that our method is robust to the amount of noise.

Interestingly, we see that our method works even without any noise (standard deviation 0.0). We suspect that this is due to the fact that half of the time in the forward computation we use the discrete values anyway and pass gradients through to the dense part. Also, note that a standard deviation of 1.5 still works, despite the fact that our saturating sigmoid is saturated for values above 2.4 as $1.2 \cdot \sigma(2.4) - 0.1 = 1.0002$. Finally, with deviation 1.0 the small model achieves DSAE of 48.5%, not much worse than the 55% achieved by the large baseline model and better than the larger baseline model with Gumbel-Softmax.

## 3.3 DECIPHERING THE LATENT CODE

Having trained the models, we try to find out whether the discrete latent symbols have any interpretable meaning. We start by asking a simpler question: do the latent symbols correspond to some fixed phrases or topics?

We first investigate this in a 32-fold compressed character-level language model. We set $c(s)$ to 4 random latent symbols $[l_1, l_2, l_3, l_4]$ and decode $s$ with beam search, obtaining:

```
All goods are subject to the Member States'
environmental and security aspects of the common
agricultural policy.
```

Now, to find out whether the second symbol in $c(s)$ stands for anything fixed, we replace the third symbol by the second one, hoping for some phrase to be repeated. Indeed, decoding $s$ from the new $c(s) = [l_1, l_2, l_2, l_4]$ with beam search we obtain:

```
All goods are charged EUR 50.00 per night and EUR
50.00 per night stay per night.
```

Note that the beginning of the sentence remained the same, as we did not change the first symbol, and we see a repetition of *EUR 50.00 per night*. Could it be that this is what that second latent symbol stands for? But there were no *EUR* in the first sentence. Let us try again, now changing the first symbol to a different one. With $c(s) = [l_5, l_2, l_2, l_4]$ the decoded $s$ is:

```
All bedrooms suited to the large suite of the large
living room suites are available.
```

We see a repetition again, but of a different phrase. So we are forced to conclude that the latent code is structured, the meaning of the latent symbols can depend on other symbols before them.

Failing to decipher the code from this model, we try again with an 8-fold compressed character-level language model that uses the *local* version of the function $c(s)$. Recall (see Section 2.3) that a local function $c(s)$ with 8-fold compression generates every latent symbol from the exact 8 symbols that correspond to it in $s$, without any context. With this simpler $c(s)$ the model has lower DSAE, 35%, but we expect the latent symbols to be more context-independent. And indeed: if we pick the first 2 latent symbols at random but fix the third, fourth and fifth to be the same, we obtain the following:

```
It's studio, rather after a gallery gallery ...
When prices or health after a gallery gallery ...
I still offer hotels at least gallery gallery ...
```

So the fixed latent symbol corresponds to the word *gallery* in various contexts. Let us now ignore context-dependence, fix the first three symbols, and randomly choose another one that we repeat after them. Here are a few sample decodes:

```
Come to earth and culturalized climate climate ...
Come together that contribution itself, itself, ...
Come to learn that countless threat this gas threat...
```

In the first two samples we see that the latent symbol corresponds to *climate* or *itself,* respectively. Note that all these words or phrases are 7-characters long (and one character for space), most probably due to the architecture of $c(s)$. But in the last sample we see a different phenomenon: the latent symbol seems to correspond to *X threat*, where *X* depends on the context, showing that this latent code also has an interesting structure.

## 3.4 MIXED SAMPLE-BEAM DECODING

From the results above we know that our discretization method works quantitatively and we see interesting patterns in the latent code. But how can we use the autoencoder models in practice? One well-known problem with autoregressive sequence models is decoding. In settings where the possible outputs are fairly restricted, such as translation, one can obtain good results with beam search. But results obtained by beam search lack diversity (Vijayakumar et al., 2016). Sampling can improve diversity, but it can introduce artifacts or even change semantics in translation. We present an example of this problem in Figure 3. We pick an English sentence from the validation set of our English-German dataset and translate it using beam search and sampling (left and middle columns).

In the left column, we show top 3 results from beam search using our baseline model (without autoencoder). It is not necessary to speak German to see that they are all very similar; the only difference between the first and the last one are the spaces before "%". Further beams are also like this, providing no real diversity.

In the middle column we show 3 results sampled from the baseline model. There is more diversity in them, but they still share most of the first half and unluckily all of them actually changed the semantics of the sentence in the second half. The part *African-Americans, who accounted however for only 13% of voters in the State* becomes *The american voters were only 13% of voters in the state* in the first case, *African-Americans, who accounted however for only 13% of all people in the State* in the second one, and *African-Americans, who elected only 13% of people in the State* in the third case. This illustrates the dangers of just sampling different words during decoding.

Using a model with access to the autoencoded part $c(s)$ presents us with another option: sample $c(s)$ and then run beam search for the sequence $s$ appropriate for that $c(s)$. In this way we do not introduce low-level artifacts from sampling, but still preserve high-level diversity. To sample $c(s)$ we train a language model on $c(s)$ with the same architecture as the model for $s$ (and also conditioned on the input), but with a different set of weights. We then use the standard multinomial sampling from this model to obtain $c(s)$ and run a beam search on the model for $s$ with the sampled $c(s)$.

In the right column in Figure 3 we show 3 samples obtained in this way. As you can see, these samples are much more diverse and they still preserve the semantics of the original sentence, even

*English sentence:*

For example, during the 2008 general election in Florida, 33% of early voters were African-Americans, who accounted however for only 13% of voters in the State.

| *Base model, beam decoding.* | *Base model, sampling.* | *Mixed decoding.* |
|---|---|---|
| Während der Parlamentswahlen 2008 in Florida beispielsweise waren 33 % der frühen Wähler Afroamerikaner, die jedoch nur 13 % der Wähler im Staat ausmachten. | So waren zum Beispiel bei den Parlamentswahlen 2008 in Florida 33 % der frühen Wähler Afroamerikaner. Die amerikanischen Wähler waren aber nur 13 % der Wähler im Staat. | Es stellte sich beispielsweise im Verlauf der Parlamentswahlen in Florida heraus, dass 33% der frühen Wähler zu den afrikanischen Amerikanern zählten, die allerdings nur 13% der Wähler des Staates betrafen. |
| Während der Parlamentswahlen 2008 in Florida beispielsweise waren 33 % der frühen Wähler Afroamerikaner, die jedoch nur 13 % der Wähler im Staat stellten. | So waren während der Parlamentswahlen 2008 in Florida 33 % der frühen Wähler Afroamerikaner, die aber nur 13 % der Bevölkerung im Staat ausmachten. | Dabei ist zum Beispiel im Laufe der Parlamentswahlen 2008 in Florida 33% in den frühen Wahlen der Afro-Amerikaner vertreten, die allerdings nur 13% der Wähler des Staates betrafen. |
| Während der Parlamentswahlen 2008 in Florida beispielsweise waren 33% der frühen Wähler Afroamerikaner, die jedoch nur 13% der Wähler im Staat ausmachten. | So waren während der Parlamentswahlen 2008 in Florida 33% der frühen Wähler Afroamerikaner, die jedoch nur 13% der Bevölkerung im Staat wählten. | 33% der frühen Wähler beispielsweise waren während der Hauptwahlen 2008 in Florida afrikanische Amerikaner, die für einen Anteil von nur 13% der Wähler im Staat verantwortlich waren. |

Figure 3: Decoding from baseline and autoencoder-enhanced sequence-to-sequence models.

if with sometimes strange syntax. One would back-translate the first example as: *In turned out, for example, in the course of the parliamentary elections in Florida, that 33% of the early voters are African-Americans, which were, however, only 13% of the voters of the state.* Note the addition of *It turned out* and restructuring of the sentence. In the third sample the whole order is reversed, as it starts with *33% of the voters ...* instead of the election phrase. Obtaining such samples that differ in phrase order and other aspects but preserve semantics has been a challenge in neural translation.

## 4  CONCLUSION

In this work, the study of text autoencoders (Bowman et al., 2016; Yang et al., 2017) is combined with the research on discrete autoencoders (Jang et al., 2016; Maddison et al., 2016). It turns out that the semantic hashing technique (Salakhutdinov & Hinton, 2009b) can be improved and then yields good results in this context. We introduce a measure of efficiency of discrete autoencoders in sequence models and show that improved semantic hashing has over $50\%$ efficiency. In some cases, we can decipher the latent code, showing that latent symbols correspond to words and phrases. On the practical side, sampling from the latent code and then running beam search allows to get valid but highly diverse samples, an important problem with beam search (Vijayakumar et al., 2016).

We leave a number of questions open for future work. How does the architecture of the function $c(s)$ affect the latent code? How can we further improve discrete sequence autoencoding efficiency? Despite remaining questions, we can already see potential applications of discrete sequence autoencoders. One is the training of multi-scale generative models end-to-end, opening a way to generating truly realistic images, audio and video. Another application is in reinforcement learning. Using latent code may allow the agents to plan in larger time scales and explore more efficiently by sampling from high-level latent actions instead of just atomic moves.

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
