# OpenReview forum: "Discrete Autoencoders for Sequence Models"
_ICLR.cc/2018/Conference — Reject_

### Official Review · AnonReviewer1 · 2017-11-26
**Discrete Autoencoders for Sequence Models**

**Rating:** 5
**Confidence:** 5

**Review:**

The topic is interesting however the description in the paper is lacking clarity. The paper is written in a procedural fashion - I first did that, then I did that and after that I did third. Having proper mathematical description and good diagrams of what you doing would have immensely helped. Another big issue is the lack of proper validation in Section 3.4. Even if you do not know what metric to use to objectively compare your approach versus baseline there are plenty of fields suffering from a similar problem yet  doing subjective evaluations, such as listening tests in speech synthesis. Given that I see only one example I can not objectively know if your model produces examples like that 'each' time so having just one example is as good as having none.

---

> ### Author Response · Authors · 2017-12-23
> **Where is a mathematical description needed more?**
>
> We thank the reviewer for the comments, but are puzzled by the sentence "Having proper mathematical description and good diagrams of what you doing would have immensely helped.". In the body of the paper, we try to give complete mathematical definitions of every term that appears there, the whole Section 2 is devoted to that, and Figures 1 and 2 are a diagramatic representation. We would like to improve them, but please give us concrete suggestions. We also double-checked the paper with external readers and it seemed to them that every single term was properly mathematically defined -- please clarify which terms are undefined to help us improve the paper. As for Section 3.4, we agree that it would be great to measure the diversity of translations in a quantitative way, not just qualitatively. But we are not aware of any metric of this kind -- we'd be happy to add the results if the reviewer can suggest one. Lack of such metric might be related to the fact that our work, to the best of our knowledge, presents the first time when an autoencoder works well enough for language to allow for such diverse results. We ask the reviewer to take this into account and possibly revise the score.

---

> > ### Comment · AnonReviewer1 · 2017-12-23
> > **Discrete Autoencoders for Sequence Models**
> >
> > You are describing a sequence model here without formally giving its mathematical equation as well as what this model depends on.  Putting that into a formal equation at the beginning of your paper makes reading immensely easier. This also enables you to introduce Figure 2 at the beginning rather than the end. Why would I want to see it at the end?
> >
> > Your treatment of auto encoding function c(s) is similar. Why not to give a block diagram to describe the process? Which parts are discrete, which parts are continuous. What would be the relation between dense(w), bottleneck(w) and c(s)? Show how does the training signal goes through these. What is the training objective function? It would have immensely helped to have a diagram of the entire process rather than drawing it myself after reading 2 pages of text.
> >
> > As I mentioned in my previous review, a single example is not meaningful and you should use metrics such as mean opinion score (MOS) used in speech synthesis. Generate N samples, ask colleagues that speak German to assess how good they are without telling them from which model these samples came from, rank the results. Find more info online.

---

> > > ### Author Response · Authors · 2018-01-05
> > > **Thank you for suggestions on how to improve the presentation**
> > >
> > > We'd like to thank the reviewer on the suggestions on how to improve the presentation. We will move Figure 2 to the beginning and add a description of a sequence model. We only omitted it because we considered it standard for ICLR, but we will move it to the front. As for the function c(s), the diagram for it is in Figure 1. The diagram for the entire process is in Figure 2, but when we move it to the front we also plan to expand it to clarify the whole process.
> > >
> > > As for using MOS, it has to our knowledge never been applied to translation and we know of no papers that would report MOS scores for WMT, so our results would be hard to compare. But even if we used a more standard metric, they all (including MOS) have the problem that they do not take the diversity into account at all. The advantage of our approach is that it can generate a diverse set of translations, but we don't know of any metric to quantify this (and we'd be grateful for pointers if one has been used before).

---

### Official Review · AnonReviewer2 · 2017-11-27
**lack of valid experiments**

**Rating:** 4
**Confidence:** 4

**Review:**

This is an interesting paper focusing on building discrete reprentations of sequence by autoencoder.
However, the experiments are too weak to demonstrate the effectiveness of using discrete representations.
The design of the experiments on language model is problematic.
There are a few interesting points about discretizing the represenations by saturating sigmoid and gumbel-softmax, but the lack of comparisons to benchmarks is a critical defect of this paper.


Generally, continuous vector representations are more powerful than discrete ones, but discreteness corresponds to some inductive biases that might help the learning of deep neural networks, which is the appealing part of discrete representations, especially the stochastic discrete representations.
However, I didn't see the intuitions behind the model that would result in its superiority to the continuous counterpart.
The proposal of DSAE might help evaluate the usage of the 'autoencoding function' c(s), but it is certainly not enough to convince people.
How is the performance if c(s) is replaced with the representations achieved from autoencoder, variational autoencoder or simply the sentence vectors produced by language model?
The qualitative evaluation on 'Deciperhing the Latent Code' is not enough either.
In addition, the language model part doesn't sound correct, because the model cheated on seeing the further before predicting the words autoregressively.
One suggestion is to change the framework to variational auto-encoder, otherwise anything related to perplexity is not correct in this case.

Overall, this paper is more suitable for the workshop track. It also needs a lot of more studies on related work.

---

> ### Author Response · Authors · 2017-12-23
> **Clarifying the comparison to dense autoencoders.**
>
> We are grateful for the reviewer for bringing up the point of comparison to dense autoencoders, such as VAEs. We've performed a number of experiments with VAEs in the same setting. The reviewer writes "How is the performance if c(s) is replaced with the representations achieved from autoencoder, variational autoencoder or simply the sentence vectors [...]". We wanted to assess this, but it is hard in principle to calculate DSAE in this case, as the number of bits in the auto-encoded representation cannot be easily measured in the dense case. In principle it's  not clear at all how to measure it for plain autoencoders or sentence vectors as even a 1-d real number can contain an unlimited number of bits (if it's a 32-bit float, it's just 32, but that's still more than our discrete representation). For VAEs, one can use the KL divergence as measure, but this is an approximate notion even in theory. In practice, all dense autoencoders, even into 4-d or 6-d vectors, begin to perfectly autoencode the sequences. So while p' is very low, it's almost impossible later to sample from those dense distributions. We tried this for VAEs as well, but even with different annealing schemes for the KL term we never managed to obtain high-quality samples, nothing comparable to our discrete results. Notably, this experience with dense autoencoders for language has been replicated by others. So our discrete method is at present the only way we know of to make autoencoders work well for language models. We hope that the reviewer will take this into account and revise the score. We are also happy to include more evaluations and will be grateful for more concrete suggestions of metrics that could be used.

---

### Official Review · AnonReviewer3 · 2017-11-30
**Autoencoders for text with a new method for using discrete latent space**

**Rating:** 6
**Confidence:** 1

**Review:**

The authors describe a method for encoding text into a discrete representation / latent space. On a measure that they propose, they should it outperforms an alternative Gumbel-Softmax method for both language modeling and NMT.

The proposed method seems effective, and the proposed DSAE metric is nice, though it’s surprising if previous papers have not used metrics similar to normalized reduction in log-ppl. The datasets considered in the experiments are also large, another plus. However, overall, the paper is difficult to read and parse, especially since low-level details are weaved together with higher-level points throughout, and are often not motivated.

The major critique would be the qualitative nature of results in the sections on “Decipering the latent code” and (to a lesser extent) “Mixed sample-beam decoding.” These two sections are simply too anecdotal, although it is nice being stepped through the reasoning for the single example considered in Section 3.3. Some quantitative or aggregate results are needed, and it should at least be straightforward to do so using human evaluation for a subset of examples for diverse decoding.

---

### Decision · Program_Chairs · 2018-01-29
**ICLR 2018 Conference Acceptance Decision**

**Decision:**

Reject

**Comment:**

This paper presents a different method for learning autoencoders with discrete hidden states (compared to recent discrete-like VAE type models). The reviewers in general like the method being proposed and are convinced that there is worth to the underlying proposal. However there are several shared complaints about the setup and writing of the paper.

- Several reviewers complained about the use of qualitative evaluation, particularly in the "Deciphering the latent code" section of the paper.
- One reviewer in particular had significant issues with the experimental setup of the paper and felt that there was insignificant quantitative evaluation, particularly using standard metrics for the task (compared to the metric introduced in the paper).
- There were further critiques about the "procedural" nature of the writing and the lack of formal justifications for the ideas introduced.